# An Aptamer That Rapidly Internalizes into Cancer Cells Utilizes the Transferrin Receptor Pathway

**DOI:** 10.3390/cancers15082301

**Published:** 2023-04-14

**Authors:** Xirui Song, Haixiang Yu, Cynthia Sullenger, Bethany Powell Gray, Amy Yan, Linsley Kelly, Bruce Sullenger

**Affiliations:** 1Department of Surgery, Duke Cancer Institute, Duke University School of Medicine, Durham, NC 27710, USA; 2Department of Pharmacology and Cancer Biology, Duke University School of Medicine, Durham, NC 27710, USA; 3Department of Biology, Duke University, Durham, NC 27710, USA; 4Department of Pharmacology and Molecular Sciences, Johns Hopkins University School of Medicine, Baltimore, MD 21205, USA

**Keywords:** aptamer, transferrin receptor, drug targeting, cancer, canine cancer

## Abstract

**Simple Summary:**

It has been a continuous effort to develop innovative targeted cancer treatments to overcome the non-specific toxicity associated with chemotherapy. Antibody-drug conjugates (ADCs) highlight the clinical benefits of specifically delivering drugs to cancer cells via targeting agents. Aptamers are high-affinity ligands that can be made synthetically and employed as targeting molecules in place of antibodies. Our group previously reported an aptamer, E3, that can target a wide range of cancer cell lines and deliver highly toxic drugs into cancer cells in vitro and in vivo. We report that E3 enters cancer cells by specifically targeting transferrin receptor 1.

**Abstract:**

Strategies to direct drugs specifically to cancer cells have been increasingly explored, and significant progress has been made toward such targeted therapy. For example, drugs have been conjugated into tumor-targeting antibodies to enable delivery directly to tumor cells. Aptamers are an attractive class of molecules for this type of drug targeting as they are high-affinity/high-specificity ligands, relatively small in size, GMP manufacturable at a large-scale, amenable to chemical conjugation, and not immunogenic. Previous work from our group revealed that an aptamer selected to internalize into human prostate cancer cells, called E3, can also target a broad range of human cancers but not normal control cells. Moreover, this E3 aptamer can deliver highly cytotoxic drugs to cancer cells as Aptamer-highly Toxic Drug Conjugates (ApTDCs) and inhibit tumor growth in vivo. Here, we evaluate its targeting mechanism and report that E3 selectively internalizes into cancer cells utilizing a pathway that involves transferrin receptor 1 (TfR 1). E3 binds to recombinant human TfR 1 with high affinity and competes with transferrin (Tf) for binding to TfR1. In addition, knockdown or knockin of human TfR1 results in a decrease or increase in E3 cell binding. Here, we reported a molecular model of E3 binding to the transferrin receptor that summarizes our findings.

## 1. Introduction

Targeted drug delivery, especially in diseases such as cancer, can ameliorate problems associated with free drug administration, including poor biodistribution, undesirable pharmacokinetics, and lack of selectivity for target tissues [1,2]. A widely appreciated strategy for targeted delivery employs antibodies as antibody-drug conjugates (ADCs) and has been extensively explored [3,4,5,6]. Aptamer-based delivery provides an exciting alternative to ADC delivery which can be challenging to manufacture and produce homogeneously [7,8,9]. Aptamers are nucleic acid high-affinity reagents generated through SELEX (Systematic Evolution of Ligands by EXponentional enrichment). They are usually 20–60 nucleotides long and, like antibodies, can bind to targets with exquisite affinity and specificity [10,11,12,13]. Unlike antibodies, aptamers are non-immunogenic, have no toxic side effects, and can be rapidly synthesized and readily conjugated with drugs, diagnostic labels, or other moieties [14,15]. Extensive chemical and pharmacological modifications allow aptamers to resist nuclease degradation and extend their half-lives in serum to several days, making them suitable for clinical administration [16,17,18,19,20]. Thus, aptamer delivery systems have great clinical potential.

We sought to select aptamers targeting prostate cancer, one of men’s most common and deadly malignancies [21]. To identify an aptamer that can specifically deliver cargo to prostate cancer cells, we previously performed cell internalization SELEX [22] against a number of different prostate cancer cell lines, including LNCaP, DU145, PC3, and 22RV1 cells [23]. The resulting aptamer, E3, binds to and internalizes into prostate cancer cells but not normal epithelial cells. By conjugating the cytotoxic drugs monomethyl auristatin E (MMAE) and monomethyl auristatin F (MMAF) to E3, we selectively kill prostate cancer cells both in vitro and in vivo. We further illustrated the significance of this aptamer as a potential delivery agent by developing antidote oligonucleotides, which could quickly and efficiently bind E3 and disrupt its function. These provide a safety switch in the event that aptamer treatment needs to be stopped or inactivated quickly.

Interestingly, although E3 was selected for prostate cancer specificity, we subsequently determined that E3 can be internalized into many additional types of cancer cells, including breast cancer, lung cancer, pancreatic cancer, skin cancer, brain cancer, peripheral blood cancer, and colorectal cancer [24]. Importantly, E3 did not significantly bind non-cancerous control cells. In addition, E3-based aptamer-highly toxic drug conjugates (ApTDCs) also show selective killing in many cancer types. This observation suggests that the target of E3 is expressed in many different types of cancers, readily internalized, and likely overexpressed on cancerous cells but not on normal cells. However, the actual target of this aptamer has not been characterized.

Previous work from our lab suggests that E3 enters cells through an endocytic pathway [23]. After surveying several cell surface markers that could be present in multiple cancers and be internalized, we found that transferrin receptor 1 (TfR1, referred to hereafter as TfR) is commonly overexpressed in many cancer cells and readily trafficked to the inside of the cells through clathrin-mediated endocytosis [25,26]. Therefore, we hypothesized that the E3 aptamer could enter cells through the TfR and determined to thoroughly validate this to better inform the most efficient therapeutic use of the E3 aptamer.

TfR is a suitable and attractive target for cancer therapy. It plays a key role in iron metabolism by mediating the internalization of iron-loading transferrins [27,28]. In malignant tissues, the expression level of TfR is much higher than in their normal tissue counterparts mainly due to their higher demand for iron [29,30,31]. TfR is also efficiently internalized, which can be exploited for the efficient delivery of therapeutic molecules [32,33]. Here, we report a carefully characterized TfR-targeting aptamer, E3, by measuring its binding on TfR recombinant protein, competing with other TfR ligands, and increasing or decreasing binding signals after knock-in and knockdown of TfR expression on cells, respectively. Based on what we learned about its targeting pattern, we proposed a molecular model of E3 binding to the transferrin receptor.

## 2. Materials and Methods

### 2.1. Cell Culture

The cell lines used in this study were grown at 37 °C with 5% CO_2_ and 95% relative humidity. In addition, 10% fetal bovine serum (Sigma-Aldrich, St. Louis, MO, USA) was added to the culture media to make “complete media” for each cancer cell line. RPMI-1640 media (Invitrogen, Waltham, MA, USA) was used to grow 22Rv1 cells (ATCC #CRL-2505, Manassas, VA, USA), and Dulbecco’s Modified Eagle Medium (Sigma, St. Louis, MO, USA) was used for Hela (ATCC #CCL-2), TLM-1 (Kerafast #EMN005-FP, Winston-Salem, NC, USA) (with 100 µg/mL Primocin), D418, B16, and B16-hTfR (with 5 µg/mL blasticidin) cells. The B16-hTfR cells were stably transduced to express human TfR (hTfR) and were kind gifts from Dr. Matthew Levy’s lab. D418 is a patient-derived xenograft osteosarcoma cell line donated by our collaborator Dr. David Hsu from Duke University.

### 2.2. Aptamer Synthesis and Labeling

Standard solid-phase nucleic acid synthesis was used for synthesizing E3 (5′-GGC UUU CGG GCU UUC GGC AAC AUC AGC CCC UCA GCC-3′), C36(5′-GGC GUA GUG AUU AUG AAU CGU GUG CUA AUA CAC GCC-3′), Waz (5′-GGG UUC UAC GAU AAA CGG UUA AUG ACC AGC UUA UGG CUG GCA GUU CCC-3′), and C2.min (5′-GGG GGA UCA AUC CAA GGG ACC CGG AAA CGC UCC CUU ACA CCC C-3′). Aptamers were synthesized with 2′fluoro modifications on the pyrimidines with an inverted dT CPG column, as described in previous publications [23]. For aptamer-dye conjugates, aptamers were synthesized with a 5′ thiol C6 S-S linker. This linker was reduced in 500 mM TCEP at 70 °C for 3 min, followed by incubation at room temperature for 1 h. Then DyLight 650 (DL650), a maleimide-activated dye (ThermoFisher, Waltham, MA, USA, #62295), was incubated with reduced aptamers at 4 °C overnight. The completion of the reaction was confirmed via analytical HPLC. Details of HPLC purification can be found by Gray et al. [23]. The unreacted maleimide reagent was removed by washing 4 times with 0.1 M NH_4_HCO_3_ through Amicon Ultracentrifugation 10K MWCO Spin Columns (MilliporeSigma, #UFC8010, Burlington, MA, USA) and stored as lyophilized pellets before use.

### 2.3. Binding Curves on Recombinant Protein and Canine Cancer Cells

Dynabeads His-Tag Isolation & Pulldown beads (Invitrogen, #10103D) were washed with SB1T buffer (40 mM HEPES, 125 mM NaCl, 5 mM KCl, 1 mM MgCl_2_, 1 mM CaCl_2_, and 0.05% Tween20) and collected through magnetic pulldown three times. The beads were then incubated with His-tagged human transferrin receptor (Sino Biology, #11020-H07H, Beijing, China) for 30 min at room temperature to saturate the binding sites on beads. After incubation, the bead-protein complexes were washed with SB1T buffer, collected through magnetic pulldown, and divided into 1.5 mL microcentrifuge tubes so that each tube contained ~2 pmole of proteins. Meanwhile, aptamer-dye conjugates were serially diluted in SB1T buffer and incubated at 65 °C for 5 min, followed by a 15 min slow cool to room temperature. Salmon sperm DNA (ssDNA, MilliporeSigma, #262012) was added to each tube at a final concentration of 1 mg/mL. Aptamer-dye conjugates/ssDNA mixes were added to the hTfR-tagged beads and incubated at 37 °C for 30 min. Samples were washed once with SB1T buffer, and the beads were collected through magnetic pulldown. The fluorescence labeling intensity of the bead-based samples was assessed on a CytoFlex flow cytometer (Beckman Coulter, Brea, CA, USA). Binding curves were generated using GraphPad Prism (San Diego, CA, USA).

### 2.4. Aptamer and Transferrin Competition Assays

22Rv1 cells were seeded in a 24-well flat bottom tissue culture treated plate at 90,000 cells/well. The cells were incubated at 37 °C, 5% CO_2,_ and 95% relative humidity for two days. On day 3, the old cell media was removed, and 180 µL 1.11 mg/mL salmon sperm DNA (MilliporeSigma) in complete media was added to each well. The plate was returned to the incubator for 1 h. Aptamers were prepared in DPBS+/+. 20 µL 20 µM each competitor aptamer (unlabeled C2min, Waz, C36, or E3) was incubated at 65 °C for 5 min and then cooled at 4 °C for 5 min to allow for refolding. Each competitor aptamer and 20 µL 10 mg/mL diferric human transferrin (Tf; ThermoFisher, #T0665) were added to designated wells and incubated for 30 min at 37 °C. Meanwhile, 2 µM E3-dye conjugate was also incubated at 65 °C for 5 min followed by 5 min on ice. 22.2 µL of refolded E3-dye conjugate was added to each well and incubated for 30 min at 37 °C. Cells were washed with DPBS without Ca^2+^ or Mg^2+^ and trypsinized using 0.05% Trypsin (Invitrogen). The trypsin was quenched by complete media, and cells were transferred to a 96-well round bottom. The cells were collected by centrifugation at 300× *g* for 5 min in a swingout rotor. The supernatant was removed, and cells were resuspended in 100 µL DPBS+/+ with 1% BSA. The fluorescence signal was assessed on a CytoFlex flow cytometer (Beckman Coulter).

B16-hTfR binding and competition assays were performed similarly. B16-hTfR cells were placed in a 96-well plate at 30,000 cells/well. The next day, old media was removed, and fresh blocked media was added. For the control cells and cells receiving DL650-C36 or DL650-E3 alone, 90 µL fresh blocked complete media was added to each well (DMEM + 10% FBS + 1 mg/mL ssDNA) for 1 h. For other cells incubated with human transferrin, 90 µL fresh blocked complete media with 1 mg/mL Tf was added for 1 h. DL650-C36 and DL650-E3 were prepared in DPBS with Ca^2+^ and Mg^2+^ at 65 °C for 5 min and cooled on ice for 5 min. Then, 10 µL aptamer solution was added into each designated well for 1 h. Each well was washed twice with PBS to collect the cells, and 100 µL trypsin was added to each well. After quenching trypsin with 100 µL complete media, cells were transferred into a U-bottom 96-well plate. Cells were washed twice with PBS before resuspending in 100 µL flow buffer (PBS + 1%BSA).

### 2.5. siRNA Knockdown

Hela cells were plated at 20,000 cells/well in a 96-well flat bottom tissue culture treated plate (ThermoFisher) in 150 µL complete media. siRNAs (IDT, PSMA, #1, 5′-CCCAACUACAUCUCAAUA-3′, #2, 5′-CAUUAAUUAUUGAGAUGU-3′; TfR, #1, 5′-AGCACUGACCAGAUAAGA-3′, #2, 5′-CCAGCAUUCUUAUCUGGU-3′) were diluted to 2 µM in OptiMEM (ThermoFisher #31985062), and HiPerFect reagent (Qiagen, #301704, Hilden, Germany) was added to the siRNA solutions, as per manufacturer’s protocol. The mixtures were incubated at room temperature for 10 min. Then, 50 µL of each transfection solution was added to designated wells. The plate was incubated at 37 °C, 5%CO_2_ and 95% relative humidity for 72 h. To measure changes in receptor binding, transfection solution was removed, and 200 nM folded DL650-aptamer conjugates were added to the cells after blocking with ssDNA, as described above. As above, the cells were then collected after trypsinization and centrifugation and resuspended in a flow buffer (1% BSA in PBS). Flow cytometry was used to measure the intensity of the fluorescence signals.

### 2.6. Molecular Modeling

The methods used for molecular modeling were reported previously [34]. The secondary structure of E3 was predicted by the mfold web server [35]. The ternary structure with the lowest free energy was then predicted by SimRNA [36] and refined via a 100-ns molecular dynamic simulation at 310 K using GROMACS with an amber14sb_OL15 force field. The resulting structure was docked with the reported crystal structure of TfR-dimer (PDBID: 3s9n) [37] using ClusPro 2.0 web server [38]. As mentioned, the models fitting with experimental findings were refined with a 100-ns molecular dynamic simulation. The final structure of the complex was drawn by averaging 1000 snapshots of the trajectory over the last 10 ns of simulation. The interaction site between E3 and TfR was analyzed using Pymol.

## 3. Results

### 3.1. Characterization of E3 Binding to Recombinant Human TfR

To test whether E3 can recognize the transferrin receptor (TfR) and bind to it, we first measured the binding affinities between E3 and recombinant human TfR (hTfR) protein. We previously reported the apparent K_D_ values of the E3 aptamer across different cancer cell lines [23,24]. To enable us to focus on any specific interactions between E3 and hTfR, we used a bead-based approach where a fixed amount of hTfR protein was immobilized on magnetic beads, followed by the addition of different amounts of DL650-labeled aptamers.

Two other hTfR aptamer ligands, C2.min and Waz, were also tested in this assay (Figure 1a) as positive controls and for comparison. These two aptamers were previously directly selected against hTfR and characterized by the Levy Lab [39,40]. In addition, a size-matched sequence previously shown to be non-reactive, C36, was used as a negative control aptamer [13,41].

As shown in Figure 1, the apparent K_D_ value of E3 was ~4 nM, indicating a high binding affinity to hTfR (Figure 1b and Table 1), which is ~2–5 fold tighter than C2.min and Waz binding to hTfR. We note that to more accurately measure the apparent K_D_ value and B_max_ of Waz, a wider range of aptamer concentrations was used (Appendix A). Although E3 had the highest K_D_, C2.min had the highest B_max_, while E3 and Waz had similar B_max_ (Table 1).

### 3.2. E3 Competes with C2.min but Not Waz for Cell Binding and Internalization

To determine where E3 binds on TfR, we performed a competition assay where E3 would compete with transferrin (Tf), the natural ligand of the hTfR (Figure 2a). In this experiment, 22Rv1 cells, a prostate cancer cell line, were preincubated with Tf for 30 min, followed by the addition of DL650-E3 aptamers. Aptamer binding was then analyzed via flow cytometry. The flow data suggest that on 22Rv1 cells, when Tf is occupying its binding sites on hTfR, E3 shows a significantly reduced binding signal. This finding indicates that E3 most likely binds to the site where Tf binds TfR on the cell surface.

We also validated this finding by competing E3 with the other hTfR-targeting aptamers, Waz and C2.min (Figure 2c). Previous studies showed that C2.min could compete with Tf for binding, while Waz is believed to bind to the apical domains of the receptor and is not displaced by Tf [39,40]. Here, 22Rv1 cells were first incubated with unlabeled E3, C2.min, Waz, or control aptamer C36, and then DL650-labeled E3 was added to the cells. As shown in Figure 2b, C36 and E3 were used as negative and positive controls, respectively. The excessive amount of C36 does not impact the DL650-E3 binding, which suggests the shifts of the signals are the results of the competition. Waz has little impact on the uptake of DL650-E3, while C2.min significantly reduces the E3 binding signal intensity. This competition study suggests that the E3 aptamer is likely binding to the transferrin binding site on the transferrin receptor.

### 3.3. hTfR Is Sufficient to Mediate E3 Binding to Cells

The E3 aptamer was selected specifically against human prostate cancer cells, and it is unknown whether it binds cancer cells from other species. To determine whether E3 could bind mouse TfR (mTfR), recombinant mTfR was immobilized on magnetic beads and incubated with DL650-labeled E3. Although antibody binding demonstrated the presence of mTfR on the beads, E3 did not crossreact with mTfR (Appendix A).

To further test whether E3 aptamer binds human TfR and not selectively to other components of the human cell surface, we exogenously expressed the human *TfR* gene in a murine cell line, B16, to create B16-hTfR cells, and checked for E3 aptamer binding on these cells. Although the binding signal is lower on these cells than on human cancer cells, the DL650-labeled E3 binding signal on B16-hTfR cells is significantly higher than that of the control sequence DL650-C36 (Figure 3). These assays were also performed with Waz as a control to confirm the expression of hTfR. Interestingly Waz appears to bind these murine cells expressing hTfR more than E3 (Figure 3) even though they appear to bind human cancer cells similarly. Furthermore, the binding signals between dye-labeled aptamers and B16-hTfR can be competed away by adding Tf (Appendix A). These studies demonstrate that expressing hTfR alone on the surface of murine cancer cells is enough to mediate E3 binding.

### 3.4. Knockdown of TfR Abolished E3 Binding to Cells

To further confirm that TfR is the main target of E3 binding, we knocked down hTfR expression on Hela cells via siRNA, as demonstrated by flow cytometry with the hTfR-targeting aptamers, Waz and C2.min (Figure 4). In addition, we included a siRNA targeting prostate-specific membrane antigen (PSMA), which is not expressed on Hela cells, as a negative control. In a previous publication [13], to verify the functionalities of C2.min and Waz, the knockdown of hTfR expression in Hela cells using this siRNA construct was confirmed via hTfR-targeting antibodies by flow cytometry. Therefore, Waz and C2.min are reported here as positive controls (Figure 4). Cells were then interrogated for E3 or C36 binding as before. As expected, for cells that received control siRNA, E3, Waz, and C2.min all show high binding signals. However, when the hTfR expression level was reduced, the binding signals of E3, Waz, and C2.min were significantly impacted and reduced, which suggests that hTfR is an essential target of E3 for entering the cells.

### 3.5. Molecular Modeling of E3 Binding to Transferrin Receptor

Our data indicate that E3 can bind to TfR and compete with transferrin for receptor binding. In addition, previous work suggests mutations in the bulge and loop areas can significantly impact E3 binding to the cancer cells [23]. Based on these findings, we used a multi-step molecular modeling strategy [34] to simulate the binding conformation of E3 and TfR. We first used SimRNA to predict the tertiary structure of E3 with the lowest free energy [36]. The aptamer structure was then equilibrated in a 100-ns molecular dynamics simulation. This refined structure was then used to perform molecular docking with the previously reported TfR structure obtained by X-ray crystallography (PDBID: 3s9n) using the online server ClusPro2 [38], which returned 30 different binding models with the lowest free energy. These models were then assessed with three criteria based on our experimental findings: (1) At least one base in each the bulge area (U4-C7, C30-C32) and the loop area (U12-C15) closely interacts with TfR (distance < 0.4 nm); (2) The binding area of E3 on TfR overlays with the TfR binding site; and (3) since full-length E3 from its original selection (before truncation to the 36 nucleotide form used here) had a longer stem, the tail of the tertiary structure of E3 (36 mer) should not clash into the protein structure. Of all 30 possible models, only two fit all three criteria. We then equilibrated both complex structures in a 100-ns molecular dynamics simulation. After this step, one model failed to meet criterium 1 and thus was excluded, leaving one binding model, which happens to be the model with the lowest free energy among 30 candidates. In the resulting model, E3 interacts with both monomers of the TfR homodimer. It occupies ~2056 Å2 of the solvent-accessible surface of the protein, significantly overlaying with the TfR-binding domain, which may explain the competitive binding between Tf and E3 to TfR (Figure 5).

Although having no binding activity to mTfR, E3 shows moderate cross-reactivity to canine cells. Flow cytometry data indicate that E3 but not control aptamer C36 bound to TLM-1, a canine melanoma cell line, and D418, patient-derived canine osteosarcoma. In addition, excessive hTf could block E3′s access to the canine TfR (Appendix A), which suggests E3 binding to canine cancer cells is TfR-dependent. This species preference of E3 could be explained via the molecular-binding model as two amino acid residues, 396F and 507E, predicted to interact with E3′s putative binding domains, was mutated in murine but not canine TfR (Figure 5).

## 4. Discussion

In this study, we report that E3, an aptamer that can target several types of cancer cells and deliver highly toxic drugs to specifically kill cancer cells, targets the human transferrin receptor (TfR).

Overexpression of the TfR on the cancer cell surface has been well documented and remains a popular target for cancer treatment. The constitutive recycling nature of the hTfR makes it a good candidate for targeted delivery. In addition, it is also an attractive target for brain cancer, where it can be utilized to cross the blood-brain barriers (BBB) [42,43].

We validated our arguments through a series of in vitro and cell-based studies. We determined that E3 can bind to the human TfR (hTfR) protein but not to the closely related mouse TfR (mTfR). On cells, E3 competes with hTfR-binding ligands, including TfR’s native ligand, transferrin (Tf), and the C2.min aptamer previously shown to bind to hTfR and also competes with Tf for binding [40]. This finding suggests that E3 binds TfR near, if not on, the Tf-binding site. Another TfR-targeting aptamer, Waz [39], did not compete with E3 for TfR binding. Waz was reported to bind to apical domains of TfR that do not directly interact with Tf.

The apparent K_D_ of E3 for TfR is slightly better than that of C2.min and approximately 5-fold tighter than Waz. However, C2.min has a higher B_max_ value and gives higher binding signals on cells, as determined by flow cytometry.

While E3 behaved similarly to C2.min and Waz in the protein binding assay and on human cancer cells, it behaved remarkably differently in human xenograft mice models. E3 and Waz have been shown to target xenograft mice bearing 22RV1 human prostate cancer cells [13,23]. However, C2.min failed to target human tumor-bearing mice [13], which indicates C2.min cannot maintain its targeting function in vivo.

To confirm that hTfR is the main target of E3 binding, we created murine melanoma B16 cells that stably express human TfR and observed strong binding of E3 to these cells but not to the parental murine cell line. As further proof of E3′s binding to hTfR, we knocked down hTfR expression in HeLa cells through siRNA treatment and showed a correlating decrease in E3 binding.

Given TfR’s wide appeal as a target for cancer treatment, E3 adds another potential targeting weapon in the arsenal against tumor growth. Importantly, we show that E3 can crossreact with canine cancer cells. This property opens the path for large animal studies for biodistribution and preclinical IND-enabling studies for E3 development as a tumor-targeting delivery vehicle. Furthermore, it should allow the development of E3-drug conjugates as novel agents to treat dogs with cancer.

To our knowledge, this paper is the first one to directly compare aptamers that have the same target but were selected through different methods and may serve to inform future cell SELEX experiments. For example, while C2.min and Waz were selected directly against TfR, E3 was agnostically selected against prostate cancer cells, yet all three target TfR with unique differences, as described above. This speaks well for the potential of SELEX to identify disease targets. Interestingly, there have been recent reports on cell-based SELEX performed in different cancer cells, and the resulting aptamers are all reported to be TfR-targeting [44,45]. This indicates that TfR is an attractive molecule for targeted therapy and an outstanding binding target for aptamers. Furthermore, several TfR-targeting molecules are currently in pre-IND development or under clinical evaluation [46,47,48], and TfR-targeting aptamers remain an interesting class of molecules to evaluate given the ease with which aptamers can be conjugated to various therapeutic agents [14,23,24,49,50,51,52,53,54,55,56,57,58,59,60,61,62,63].

However, concerns with healthy cells expressing a high level of TfR need to be addressed. Ongoing efforts in the lab also focus on selecting an aptamer that can cross-react with mTfR for animal studies. Although dogs can be used for pre-clinical evaluation, evaluating the pharmacodynamic and pharmacokinetic features of TfR-targeting aptamers in mice would be more cost-effective.

## 5. Conclusions

TfR is an attractive target for targeted delivery in cancer treatment. In this paper, we showed that E3 competes with Tf for binding, which can further limit cancer cell growth by reducing the bioavailability of iron.

## 6. Patents

Duke University (B.P.G. and B.A.S.) has submitted patent applications on aptamer-drug conjugates.

## Figures and Tables

**Figure 1 cancers-15-02301-f001:**
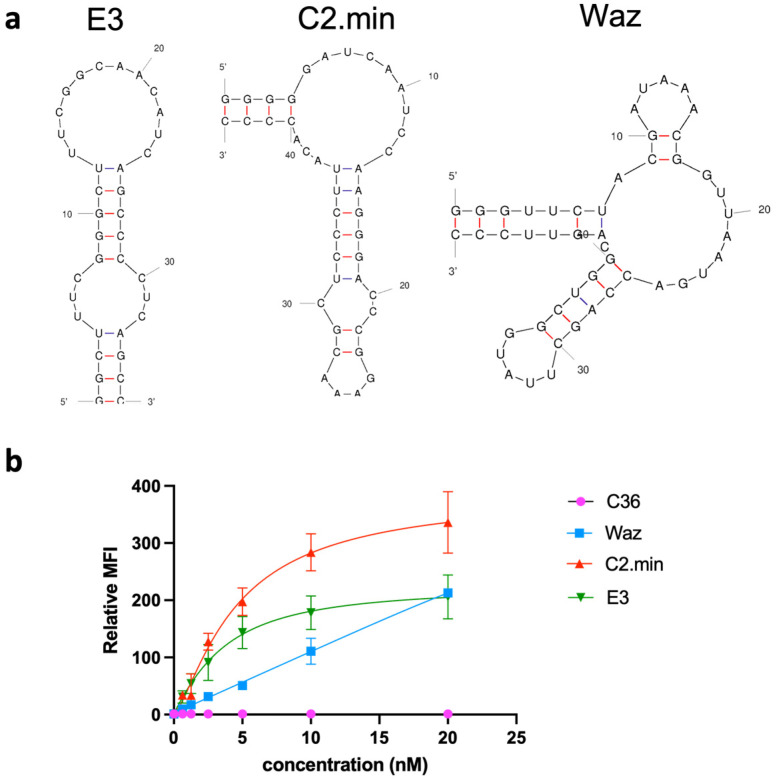
Aptamer structures and their binding on immobilized recombinant human transferrin receptor. (**a**) Predicted secondary structures of E3, C2.min, and Waz based on their sequences. Structures were generated by mFold. (**b**) A plot of median fluorescence measured via flow cytometry. Fixed amounts of immobilized hTfR were incubated with increasing concentrations of DL650-E3, DL650-C36, DL650-Waz, or DL650-C2.min. For 1 h before washing. MFI = median fluorescent intensity and signals were normalized to signals collected from the bead-TfR sample.

**Figure 2 cancers-15-02301-f002:**
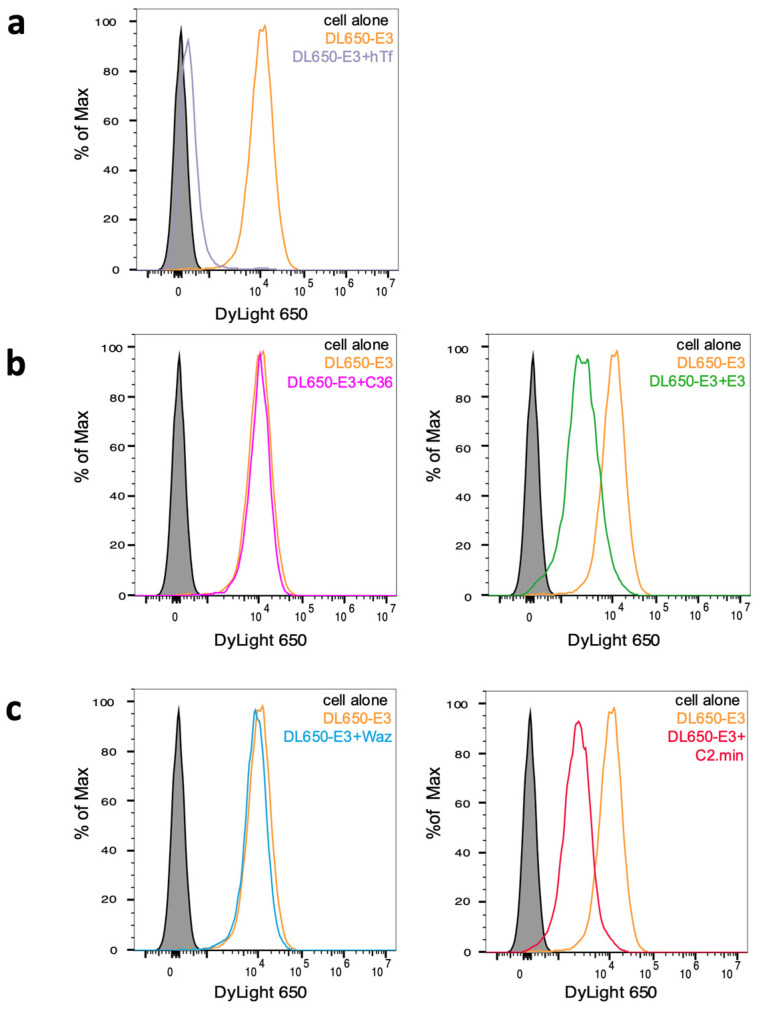
The changes of 200nM DL650-E3 binding signal in competition assays. Competitors were preincubated with 22RV1 cells. (**a**) The binding signals of DL650-E3 after competing with 1mg/mL transferrin is presented here in purple. The non-competed DL650-E3 binding signal is shown in orange. The fluorescence intensity of DL650-E3 was collected after cells were pre-incubated with a 10-fold excessive amount of (**b**) C36, itself, (**c**) Waz, and C2.min, which are shown in magenta, green, blue, and red, respectively.

**Figure 3 cancers-15-02301-f003:**
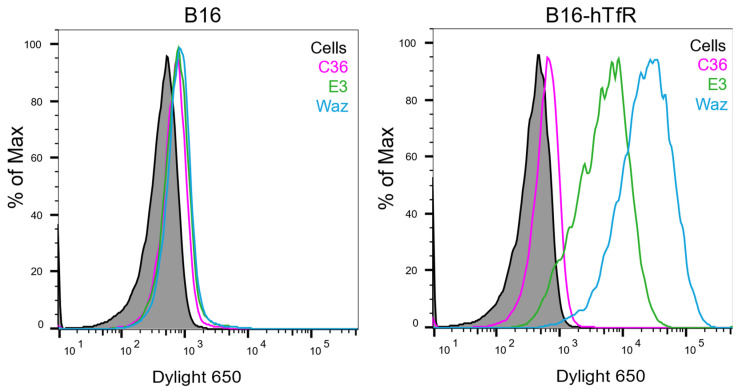
Binding signals on B16 (**left**) and hTfR-expressing B16 cells (**right**). The parental cell line (B16) was a murine melanoma cell line.

**Figure 4 cancers-15-02301-f004:**
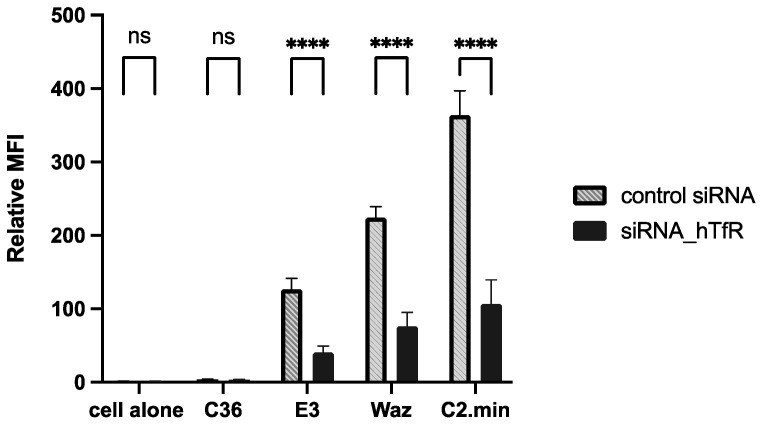
Analysis of binding signal after TfR knockdown via siRNA. The uptake of different aptamers was measured in Hela cells via flow cytometry at final concentrations of 200 nM. All data are normalized to the cell-alone signal. **** indicates *p* < 0.0001, ns = not significant, *p* > 0.05.

**Figure 5 cancers-15-02301-f005:**
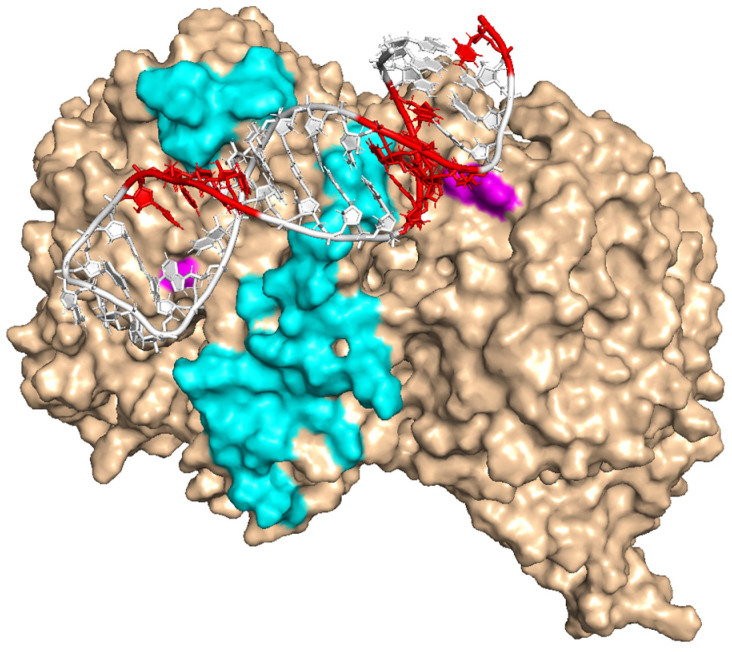
Predicted structure of E3 binding to hTfR dimer by molecular modeling. The TfR dimer is colored beige, with the Tf-binding domain highlighted in cyan, and murine mutation sites 396F and 507E are highlighted in magenta. E3 is colored white, with the putative binding bulge and loop region highlighted in red.

**Table 1 cancers-15-02301-t001:** Binding constants from data collected in Figure 1b and Appendix A.

Aptamer	Apparent K_D_ (95% CI) (nM)	B_max_ (95% CI)
E3	4.23 (3.55–5.04)	197.0 (190.4–203.9)
C2.min	7.44 (4.46–13.06)	355.3 (319.8–395.7)
Waz	22.09 (16.05–30.62)	198.8 (177.8–224.4)

## Data Availability

All data are shown within the paper.

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
