# Peer review of "An Aptamer That Rapidly Internalizes into Cancer Cells Utilizes the Transferrin Receptor Pathway"

_cancers, 2023, doi:10.3390/cancers15082301_

Round 1
Reviewer 1 Report
Major comments:
Comment1
In this manuscript, the authors have verified the binding of the E3 aptamer to TfR using the recombinant protein and TfR-expressing cells, but there is a discrepancy between the two results. The binding analysis using the TfR recombinant protein in Figure 1 shows that the E3 aptamer binds more strongly than the other two aptamers, C2.min and Waz. However, in subsequent experiments with cells in Fig. 4, E3 has a smaller MFI value compared to the other two aptamers. Could this be due to the fact that the recombinant protein fails to reflect native TfR protein? And, based on the results of Fig. 4, I suspect that the E3 aptamer may be inferior to the other two in binding to native TfR protein. If so, the novelty is questionable, including the fact that the binding region is similar to that of C2.min.
Comment2
The authors focused only on binding, but have the authors verified whether the ApTDC activity is stronger than the other TfR aptamers? I would guess that this may be important in demonstrating the usefulness of this aptamer.
Minor comments:
・In the Material Methods section, the authors should also write the catalog number of the reagent the authors purchased.
・In several histograms, the authors have used the term “Normalized to mode”, but it seems the authors mean “Cell Count (Normalized)”. It is better to change it.
・In the experiment in Figure 4, have the authors ever checked the efficiency of knockdown by the siRNA using western blot or other methods?
・In the experiment in Figure 5, to examine the potential for the animal model, the authors have verified only the binding to dog cancer cells. I am wondering it is sufficient and useful. Have the authors ever checked for binding to non-cancer cells? It would be important to show that the E3 aptamer is likely to accumulate in cancer cells and unlikely to accumulate in non-cancer cells, using dog cells.
・.I wonder if the authors have ever verified binding with canine TfR. And it is necessary to verify that the binding of this aptamer to canine cancer cells is TfR-dependent using siRNA.
Reviewer 2 Report
This paper reports that an aptamer E3 selectively internalizes into cancer cells utilizing a pathway that involves transferrin receptor 1 (TfR 1). E3 binds to recombinant human TfR 1 with high affinity and competes with transferrin (Tf) for binding to TfR1. Knockdown or knockin of TfR1 results in a decrease or increase in E3 cell binding, respectively. E3 does recognize canine cancer cells, but does not bind to murine cancer cells, providing a translational development path for potentially treating canine and human cancers. The results are very significant for authors, and the manuscript is well composed. It’s recommended this paper to be published in Cancers after minor revision as following:
1. It’s suggested to delete “)” of “a)” etc. in all figures.
2. “---cancer delivery target” in “5. Conclusions” part is not easy understood, it’s suggested to delete “delivery”.
Reviewer 3 Report
In this manuscript, the authors report an aptamer, E3, that can target and enter a number of cancer cells across species by binding to the human transferrin receptor (TfR). With competitive assay, siRNA know down assay and molecular modeling assay, the targeting site on TfR of E3 is confirmed and the specificity of E3 towards hTfR is established on several cancerous cell lines. I believe majority of the arguments in the manuscript are well supported by the data provided in the manuscript. The study is important for the translational clinical applications of E3 in TfR-based carcinoma treatments, though the current study does rule out the possibility of using E3 on xenografted mouse model for pre-clinical tests. However, the experiments on canine cell lines are quite detached from the scope of this manuscript, though the experiments seem to give a positive and promising outcome. I understand that the authors may want to boost up the confidence in E3 for large animal studies and further translational applications. The part on canine cell lines could be reported in a separate manuscript after the authors have a better understanding on the E3 behavior with canine TfRs. I would recommend the manuscript for publication after the authors have revised the current manuscript according to the comments.
Round 2
Reviewer 1 Report
Revisions are fine to this referee and the manuscript is now acceptable for publication.